# Enhanced Natural Attenuation of Groundwater Cr(VI) Pollution Using Electron Donors: Yeast Extract vs. Polyhydroxybutyrate

**DOI:** 10.3390/ijerph19159622

**Published:** 2022-08-04

**Authors:** Marina Tumolo, Angela Volpe, Natalia Leone, Pietro Cotugno, Domenico De Paola, Daniela Losacco, Vito Locaputo, Maria Concetta de Pinto, Vito Felice Uricchio, Valeria Ancona

**Affiliations:** 1Water Research Institute, Italian National Research Council (IRSA-CNR), 70132 Bari, BA, Italy; 2Department of Biology, University of Bari, 70126 Bari, BA, Italy; 3Department of Chemistry, University of Bari, 70126 Bari, BA, Italy; 4Institute of Biosciences and Bioresources, Italian National Research Council (IBBR-CNR), 70126 Bari, BA, Italy

**Keywords:** hexavalent chromium, groundwater, bioremediation, enhanced natural attenuation, yeast extract, polyhydroxybutyrate

## Abstract

Remediation interventions based on the native bacteria’s capability to reduce Cr(VI) represent a valid strategy in terms of economic and environmental sustainability. In this study, a bioremediation test was carried out using viable microcosms set with groundwater and deep soil (4:1), collected from the saturated zone of an industrial site in Southern Italy that was polluted by ~130 µg L^−1^ of Cr(VI). Conditions simulating the potential *natural attenuation* were compared to the *enhanced natural attenuation* induced by supplying yeast extract or polyhydroxybutyrate. Sterile controls were set up to study the possible Cr(VI) abiotic reduction. No pollution attenuation was detected in the unamended viable reactors, whereas yeast extract provided the complete Cr(VI) removal in 7 days, and polyhydroxybutyrate allowed ~70% pollutant removal after 21 days. The incomplete abiotic removal of Cr(VI) was observed in sterile reactors amended with yeast extract, thus suggesting the essential role of native bacteria in Cr(VI) remediation. This was in accordance with the results of Pearson’s coefficient test, which revealed that Cr(VI) removal was positively correlated with microbial proliferation (n = 0.724), and also negatively correlated with pH (n = −0.646), dissolved oxygen (n = −0.828) and nitrate (n = −0.940). The relationships between the Cr(VI) removal and other monitored parameters were investigated by principal component analysis, which explained 76.71% of the total variance.

## 1. Introduction

Chromium is a transition metal with complex chemistry. Its oxidation states vary between +6 and −2 depending on several physical-chemical parameters, including the pH and redox potential [1]. The hexavalent form of chromium is responsible for environmental and health concerns because it is highly soluble and spreads through the environmental matrices and is toxic to living organisms. Cr(VI) in groundwater at concentrations above 70 µg L^−1^ is mainly related to the improper disposal of waste and wastewaters resulting from industrial activities [2]. This represents a global environmental problem. In fact, Cr(VI) is involved in a number of industrial applications, including metal plating, leather tanning, the synthesis of paint pigments and dyes, and stainless-steel and refractory production [3].

Remediation interventions are aimed at reducing Cr(VI) to the less toxic and sparing soluble Cr(III). In fact, the main Cr(III) behavior in water is the formation of chromium hydroxides, which tend to precipitate under neutral and alkaline conditions [4]. Common Cr(VI) reduction strategies are based upon physical-chemical treatments, both ex situ and in situ. As an alternative, bioremediation (bioreduction) can be convenient in terms of economic and environmental sustainability.

In certain sites, pollution caused by Cr(VI) may be reduced without or with little human intervention by *natural attenuation*, thanks to naturally occurring electron donors, i.e., aqueous Fe(II), Fe(II)-bearing minerals, nitrates, sulphates and organic compounds [5,6,7]. Both aerobic and anaerobic Cr(VI) bioreduction have been observed at the intracellular or extracellular level as part of a defense mechanism in response to oxidative stress. Otherwise, chromate may be utilized as a terminal electron acceptor in anaerobic respiration by *dissimilatory metal-reducing bacteria (DMRB)*, including dissimilatory iron-reducing bacteria [5,8,9]. In addition, indirect Cr(VI) reduction may occur coupled with nitrate and sulphate bacterial reduction [10,11,12,13].

Although the interest in Cr(VI) bioremediation has visibly increased in the field of heavy metal remediation, the feasibility of this kind of intervention has some limitations. In particular, most bacteria capable of Cr(VI) reduction are heterotrophic, and nutrients are a limiting factor in oligotrophic environments, such as groundwater [14]. To overcome this issue, a promising approach is to supply organic amendments to the aquifer system for supporting both the native microorganisms’ proliferation and Cr(VI) bioreductive processes, thus enhancing, or even inducing, the natural attenuation processes mediated by native bacteria [7,15,16,17]. This remediation strategy can be defined as *enhanced natural attenuation* [18,19,20].

Carlos et al. [21] observed that the addition of low-cost nutrients, including sucrose and urea, to reactors supported bacteria isolated from an industrial landfill effluent in reducing 4 mg/L of Cr(VI) within 72 h of incubation at 30 °C. Likewise, a microcosm experiment by Lin et al. [22] demonstrated that, starting with 40 mg/L of Cr(VI), a complete chromium reduction via biological mechanisms can be obtained within 80 days using cane molasses as a carbon source under anaerobic conditions. Moreover, recent studies demonstrated that the use of yeast extract, as a carbon source and electron donor, positively affects Cr(VI) bioremediation [14,23]. Aulenta et al. [24] and Baric et al. [25] demonstrated the efficacy of polyhydroxybutyrate as a slow-release electron donor in bioremediation treatments of polluted groundwater.

Studies on Cr(VI) bioreduction mostly focus on specific microorganisms that are isolated from polluted matrices and then cultivated. Nevertheless, in natural groundwater systems, the reduction of chromate is not conducted by a single microbe; rather, it depends on a mixed microbial community, which interacts with the environment surrounding the cells [26]. 

This study aimed to evaluate the feasibility of a natural attenuation or, alternatively, an enhanced natural attenuation strategy for recovering Cr(VI)-polluted groundwater in the industrial site of the Barletta Municipality in Southern Italy. For practical reasons, studies on aquifer remediation focus in general on groundwater samples, while deep saturated soil collection is not included due to the complex tasks required, i.e., drilling operations [27]. However, there are numerous interactions between groundwater and deep saturated soil involving their microbial and mineral components. In light of this aspect, a series of water:deep soil (4:1) microcosms were set in the present study. The main objectives were: (1) to study, on the microcosm scale, the capabilities of autochthonous bacteria in favoring the bioreduction of Cr(VI), both in the absence and in presence of two different amendments (yeast extract or polyhydroxybutyrate); (2) to evaluate the possible contributions of chemical reductive processes triggered by the environmental matrices’ mineral composition or the amendments supplied by using sterile controls; and (3) to investigate significant correlations between Cr(VI) removal and microbial proliferation and the physical-chemical parameters.

## 2. Study Area

The territory of Barletta, Apulia, Southern Italy, is characterized by a substrate of calcareous rock, known as the “Apulia Carbonate Platform”, which is attributable to the formation of the “Limestone of Bari” of the middle-upper Cretaceous. Deposits of “Calcarenite of Gravina” with different grain sizes, locally called “tuff”, and “Sub-Apennine Clays” rest on the substrate of the calcareous rock at 35 m below ground level. The upper stratigraphic levels consist of fine-grained quartz sand and sandstone containing subordinate intercalations of loamy sand and sandy loam sediments [28].

In accordance with the stratigraphy, previous studies reported a multilayer groundwater system with two aquifers [29,30]. The deepest one is the “Deep Apulian Aquifer auct.”, a karst aquifer characterized by the highest flow rate circulating under pressure at more than 30 m below ground level. The upper aquifer flows slightly pressurized through the sandy layers. The piezometric levels of the shallow aquifer respond to the seasonal rainfalls, and the flow direction is mainly SW–NE, towards the Adriatic coast [29]. As a shallow aquifer, it is particularly vulnerable to pollution from human activities.

The study area (Figure 1a) belongs to the industrial area of Barletta (41°18′44.6″ N 16°18′05.6″ E), where several factories have been registered since the nineteenth century. Two critical aspects characterize this site. It is surrounded by a residential area and extends along the Adriatic coast in a south–east direction for 7 km, close to the coast. In support of these considerations regarding the critical qualities of the study area, a recent study by Di Ciaula [31] demonstrated the bioaccumulation of Hg, Ni, Cd and As in children living nearby.

Since 2016, this site has been subjected to monitoring campaigns which demonstrated the groundwater’s Cr(VI) contamination risk, with a concentration of about 100 µg L^−1^, as confirmed also in the 2020 monitoring campaign, which is against the Italian legal limit of 5 µg L^−1^ [30]. The groundwater’s monitoring network, prepared by the local authorities, consists of n.41 piezometers (Figure 1b), n.21 of which were installed in 2020 in order to increase the number of monitoring wells where the contamination risk was higher.

## 3. Methodology

### 3.1. Sampling of the Deep Saturated Soil and Groundwater

Deep saturated soil samples, representative of the saturated zone, were collected on February in 2020 from new drilling near the GWG monitoring well. The drilling reached −30 m, and the saturated zone was detected at −7 m. Deep saturated soil was collected from the soil cores between depths −9 m and −27 m, transferred in glass jars with screw caps, and transported to the laboratory under refrigerated conditions. 

Groundwater samples were collected in September 2021 from the pre-existing monitoring wells GWG (41°19′02.2″ N 16°17′41.3″ E) and GWQ (41°18′54.5″ N 16°18′41.5″ E). The selection of the wells for water sampling was based on the Cr(VI) concentration, in accordance with previous monitoring campaigns funded by the municipality in 2016 and 2020 [30]. Specifically, GWG was located in a sub-area with high levels of Cr(VI), marked in red in Figure 1b. On the contrary, GWQ was in a sub-area not polluted by Cr(VI) (<0.5 µg/L) and was used in this study as a blank control (Section 3.3). The depth of the wells was 20 m, allowing for the sampling of the shallow aquifer. Taking into account the water level, measured using a tape measure, a submersible well pump was positioned at −14 m to avoid sampling at the water interface and the well bottom. The piezometers were purged at a low flow rate until the water stabilization occurred, according to Shilling et al. [32]. A volume of 30 L of water per well was collected in polyethylene (PE) tanks and stored under refrigerated conditions until its employment in the microcosm setup. Water samples for dissolved organic carbon (DOC) analysis were filtered onto a 0.2 µm polyethersulfone membrane on the field and collected in sterile 50 mL polypropylene tubes. For the total dissolved Cr and Fe determination, a volume of 100 mL of water was passed through a 0.2 µm polytetrafluoroethylene (PTFE) filter, collected in PE bottles, and acidified with ultrapure HNO_3_ up to a concentration of 1%. A water sample filtered at 0.2 µm (PTFE membrane) was collected for the Cr(VI) analysis in a 100 mL PE bottle and processed upon arrival at the laboratory. Samples for the determination of nitrate and sulphate content in water were filtered on a 0.22 µm PTFE membrane and transferred into 100 mL PE bottles. Water samples for DNA extraction were collected in sterile 1 L PE bottles, in triplicate.

All water samples were kept refrigerated until their arrival at the laboratory and stored in the dark at +4 °C until the time of the analyses. 

### 3.2. Characterization of the Environmental Matrices

Continuous field measurements of the groundwater physical-chemical properties, i.e., the electrical conductivity (EC), temperature, pH, dissolved oxygen (DO) and redox potential (ORP), were performed using an immersed multiparameter tester. Total dissolved Cr and Fe were determined in the groundwater samples from the GWG and GWQ wells by inductively coupled plasma mass spectrometry analyses (ICP-MS, Agilent 7700 Series ICP-MS, Agilent Technologies, Tokyo, Japan). The dissolved Cr(VI) content was determined within 24 h of the water sampling by using the 1,5-diphenylcarbazide method [33]. The absorbance was read at 540 nm using a UV–Vis spectrophotometer (Cary 60-UV-Vis, Agilent Technologies, Santa Clara, CA, USA) equipped with a cell with optical path lengths of 10 cm to enhance the spectrophotometric detection, according to Ancona et al. [14]. A matrix-matched calibration curve was obtained using the Cary WinUV 5.0 Software by spiking aliquots of the GWQ water, previously kept in contact with the sampled soil with potassium dichromate at known concentrations.

The nitrate and sulphate contents were determined in the groundwater samples by ionic chromatography (Metrohm 930 compact IC flex 508, Metrohm Nederland, Barendrecht, The Netherlands).

The deep saturated soil properties, including the pH, water content, texture (according to the USDA textural classes), cation-exchange capacity (CEC), soil electrical conductivity (EC), organic carbon (OC) and carbonate content (CC), were determined in accordance with the Italian Official Methods of Soil Chemistry (MUACS), approved by the Minister for Agricultural Policies [34]. In addition, the total Cr and Fe in the deep saturated soil were measured by ICP-MS after the microwave digestion of 0.5 g of finely-ground DS with HCl and HNO_3_ (3:1 *v*/*v*) for 15 min at 200 °C.

Genomic DNA (eDNA) was extracted from the groundwater and deep saturated soil, for the purpose of setting up the microcosm tests, using a DNeasy PowerSoil kit (Qiagen, Germantown, MD, USA) coupled with the semi-automatic extractor Qiacube (Qiagen, Germantown, MD, USA). The protocol performed was the IRT^®^, the Inhibitory Removal Technology^®^, which is designed to remove various inhibitors, and the elution volume was set at 50 µL for concentrating the microbial genomic DNA. For the water samples from the wells, one liter was filtered through a 0.22 µm PC membrane, which was treated as the solid matrix. For the deep saturated soil, eDNA was extracted from 0.25 g, according to the manufacturer’s instructions. 

The integrity of the extracted eDNA was verified on 1% agarose gel through electrophoresis. The eDNA concentration was quantified by spectrofluorometric assays using Qubit 3.0 coupled with the High Sensitive Qubit dsDNA assay kit (Thermo Fisher Scientific, Waltham, MA, USA).

### 3.3. Experimental Design and Microcosm Setup 

A batch experiment was carried out using environmental matrices from the study area to evaluate the capability of Cr(VI) natural attenuation compared to enhanced natural attenuation processes using two different electron donors: yeast extract and polyhydroxybutyrate. Before the microcosm setup, the deep saturated soil was previously air dried, pestled and sieved through a 2 mm sieve.

The experimental design included 12 series of reactors (Table 1), set up in 250 mL PE bottles each containing 200 mL of groundwater and 50 g of dry soil (DS). The sterile microcosms were prepared under a microbiological hood using previously autoclaved (20′ at 121 °C and 1 bar) water, soil and amendments, and bottles previously sterilized under UV light for 30 min.

The BIO, YE and PHB series, as well as the corresponding sterile controls ABIO, ABIO YE and ABIO PHB, were prepared with contaminated water from the GWG well.

The BIO series was intended to study the Cr(VI) natural attenuation.

The YE series had the purpose of evaluating the Cr(VI) bioreduction enhanced by yeast extract as electron donor. A 50 g/L solution of yeast extract (Liofilchem srl, Roseto d. Abruzzi, Italy) was prepared, and aliquots of 800 µL were added to obtain a final concentration of 200 mg/L of amendment per bottle.

To evaluate the Cr(VI) microbial reduction promoted by polyhydroxybutyrate, the PHB series was set up by adding 36 mg per bottle of polyhydroxybutyrate powder (max particle size 300 µm, Goodfellow Cambridge Ltd., Huntingdon, UK) to obtain a final concentration of 180 mg/L.

The sterile series, ABIO, ABIO YE, and ABIO PHB, were set up at the end of the live-cell experiments to evaluate the possible occurrence of chemical reduction. 

Blank microcosms (blk) containing DS and water not polluted by chromium, sampled from the GWQ piezometer, were set up for each of the above treatments and used as system blanks in the spectrophotometric measurement of Cr(VI).

All microcosms were incubated in the dark at 21.6 °C, which corresponds to the field-measured aquifer temperature (Section 4.1), and manually shaken two times per day.

Microcosms were sacrificed in triplicate after fixed time spans for the analysis. 

### 3.4. Microcosms’ Monitored Parameters

The analyses described below were carried out in the aqueous phase of each microcosm at fixed monitoring times. Six times of investigation were chosen for the monitoring of the viable reactors (BIO, YE and PHB series), corresponding to 2, 4, 7, 9, 11 and 21 days after the microcosm setup. For the sterile reactors (ABIO, ABIO YE and ABIO PHB series), physical-chemical parameters were monitored at three times of investigation, corresponding to the 4th, 11th and 21st reaction days.

The water fraction was sampled using sterile syringes, immediately filtered on 0.22 µm polycarbonate membranes, and stored at −20 °C for further analysis. Moreover, the membranes were stored at −20 °C and processed for the eDNA extraction. 

The dissolved Cr(VI), nitrate and sulphate were determined as previously described in Section 3.2. 

Other monitoring parameters were pH, ORP and DO, respectively measured by using the pH/ORP meter (HI9025 Hanna Instrument, Rhode Island, USA) and the dissolved oxygen meter (HI9143 Hanna Instrument, Hwy Smithfield, RI, USA).

In addition, for the viable batches only, belonging to the BIO, YE and PHB series, eDNA was extracted. In particular, 50 mL of water from each microcosm was sampled using a sterile syringe and filtered onto a 0.22 µm membrane. The membranes were then treated as described in Section 3.2.

### 3.5. Statistical Analyses

Statistical analyses were performed using the XLSTAT software (version 2022.2.1) for evaluating significant correlations between the monitored parameters and differences between the treatments (unamended, amended with yeast extract and amended with polyhydroxybutyrate) applied in both the presence and the absence of the native bacteria. The principal component analysis (PCA) and cluster analysis (CA) were performed using a dataset built with the values of the Cr(VI) removal, pH, ORP, DO, eDNA, nitrate and sulphate, which were measured at the end of the experiment, i.e., after 21 days, for all microcosms except for the YE set. For these microcosms, the point of maximum pollutant removal, i.e., 7 days, was considered as the end of the experiment.

Ecological issues must be properly considered as the combined effects of multiple variables which interact with one another. According to this concept, in ecological experiments, in statistical analyses, a multivariate analysis should be preferred over multiple univariate analyses. In fact, the differences among groups can be affected not by one variable alone, but rather by the entire set of variables [36]. In the light of this, a multivariate analysis of variance (MANOVA) was also carried out on the above-mentioned dataset.

#### 3.5.1. PCA and CA

The PCA performed on the above-mentioned dataset was used to simultaneously analyze the microcosms’ monitored parameters and the relationships between them. According to the PCA scores, the variables were plotted in a two-dimensional graph representing the space of the first two principal components, which explain a significant proportion of the data variance. The PCA scores were also used for a cluster analysis (CA). The agglomerative hierarchical clustering (AHC) and the k-means clustering approaches were tested for building clusters in which the variance within the groups is minimized, while the variance between groups is maximized.

#### 3.5.2. MANOVA

To evaluate the effects of the different experimental setups on the investigated properties, a multivariate analysis of variance (MANOVA) was performed. The MANOVA analyses the differences among groups, considering the intercorrelations of the independent variables. The Wilk’s test, Hotelling–Lawley test, Pillai test and Roy test of significance were applied to evaluate the statistical differences between the groups. Usually, Pillai’s trace has the highest statistical power. When the significance of the MANOVA test was verified, the univariate ANOVA test was performed to determine which variables and factors influenced the significance.

## 4. Results

### 4.1. Characterization of the Environmental Matrices

The main physical-chemical properties, nitrate, sulphate, DOC, Fe, Cr and Cr(VI) content of the groundwater and deep saturated soil samples used for the microcosm setup are listed in Table 2.

Values exceeding the prescribed limit were found in the GWG groundwater sample for total chromium and Cr(VI). It should be specified that the analytical techniques used for the determination of the total Cr and Cr(VI) in water have different accuracies, and this explains the slight difference between the two parameters in the GWG. However, these results demonstrated that the total chromium in this sample was entirely in the form of hexavalent chromium, and both exceeded the legal limit. No detectable Cr(VI) was found in the GWQ, validating the possibility of using the GWQ water to set up the blank microcosms. 

The total Cr content of the sampled soil was in line with the world average values for sandstones (20–40 mg/kg) and calcareous soils (5–16 mg/kg) [37]. By contrast, the total Fe content was remarkable.

With regards to the total DNA extracted from the matrices, unquantifiable genomic material was extracted from the deep saturated soil, indicating its very low content of bacterial cells. For this reason, investigations of the microbial proliferation in the microcosms focused on the aqueous phase. For the groundwater samples, the mean values measured in the GWG and GWQ wells were, respectively, 1.69 ± 0.6 and 1.49 ± 0.1 ng/µL. Electrophoresis on 1% agarose gel confirmed the eDNA integrity.

### 4.2. Microcosm Monitoring 

The percentage of Cr(VI) removal and the eDNA content observed during the experiment in the viable reactors are shown in Figure 2.

The YE exhibited the quickest reduction of Cr(VI), reaching 100% Cr(VI) removal after 7 days of incubation. Concurrently, a rapid and consistent increase in the eDNA yield was measured in YE already after 2 days, with stable values until the ninth day, followed by a rapid decrease. A slower, but relevant, pollutant removal was also observed in the PHB, starting from the seventh day, and reaching about 70% at the end of the experiment. Microbial proliferation in the PHB was also lower in comparison to the YE series, and gradually increased after 7 days. Neither evidence of natural attenuation nor evidence of microbial proliferation were observed in the unamended microcosms (BIO).

The values of the Cr(VI) removal (%) in the sterile reactors are plotted in Figure 3.

The abiotic removal of Cr(VI) was observed in the ABIO YE. In particular, the highest removal rate occurred in the first four days and then proceeded slowly up to a maximum value of about 67% after 11 days. By contrast, no pollutant removal was observed in the ABIO or ABIO PHB series.

The mean values of the pH, DO, ORP, nitrate, sulphate and eDNA measured at fixed monitoring times in the viable and sterile reactors are shown in Table 3.

Comparing the viable series, BIO demonstrated the lowest decrease in pH and DO during the experiment (from 7.3 ± 0.02 to 7.2 ± 0.02 and from 5.4 ± 0.13 to 4.3 ± 0.16 ppm, respectively). ORP varied slowly without dropping below values equal to 199 ± 12.76 mV until the end of the experiment. Interestingly, BIO demonstrated the highest increase in nitrate and sulphate content until the fourth day of incubation (from 50.40 ± 3.55 to 100.60 ± 4.00 mg/L and from 481.18 ± 2.61 to 1242.70 ± 50.90 mg/L, respectively), which then gradually decreased to values closer to initial nitrate and sulphate content.

In the YE, the most remarkable feature was the zeroing of the DO and nitrate after two days of incubation. In parallel, ORP also decreased drastically from 161.7 to 1.16 mV. DO and ORP gradually increased from the fourth day to the end of the experiment, while the nitrate content remained below the instrumental detection limit until the end. The pH values varied from ~7.2 to ~7.0, as measured on the 21st day. As previously described for the BIO series, in YE the sulphate content also notably increased until the fourth day, from 535.68 ± 22.23 to 890.30 ± 56.21 mg/L, and then gradually decreased to a value closer to initial one at the end of the experiment.

In PHB, the pH started to decrease gradually after the 7th day, from ~7.3 to ~7.0. The DO varied from 5.1 ± 0.07 to 1.1 ± 0.09 ppm, decreasing gradually throughout the experiment. In this treatment, variations in the ORP values did not describe a clear increasing or decreasing trend. By contrast, as previously described for the BIO and YE, in PHB an increasing trend was also observed at the earliest monitoring times for nitrate and sulphate. In particular, the nitrate content increased from 50.30 ± 4.23 to 71.35 ± 0.55 mg/L by the seventh day and decreased to values below the instrumental detection limit at the two latest monitoring times. The highest value of sulphate content (953.70 mg/L) was achieved after 7 days of incubation and decreased to ~600 mg/L at the following incubation times.

With regards to the sterile series, ABIO, ABIO YE and ABIO PHB, it should be noted that the starting values of the pH and sulphate were higher than in the viable series, possibly as an effect of the intracellular compound release after the bacterial death. Overall pH values at the end of the experiment were lower than those at the earlier monitoring times. This aspect was most evident in the ABIO YE (starting value ~7.9 and a value of ~7.4 at the 21st day). Other parameters which, interestingly, varied in ABIO YE were the DO and nitrate. In particular, the lowest DO content was measured at the 4th day (2.9 ± 0.06 ppm). At the same monitoring time, the nitrate was completely consumed (value was below the detection limit).

### 4.3. Statistical Analyses

#### 4.3.1. PCA and CA

In accordance with the Pearson coefficient values reported in Table 4, correlations between the pairs of variables were verified. This was a preparatory condition for the PCA application to the dataset.

The percentage of Cr(VI) removal is negatively correlated with pH (n = −0.646), DO (n = −0.828) and nitrate (n = −0.940), while it is positively correlated with eDNA (n = 0.724). The pH is positively correlated with DO (n = 0.886), nitrate (n = 0.690) and sulphate (n = 0.577). The DO is positively correlated with nitrate (n = 0.810) and negatively correlated with eDNA (n = −0.544). All other variables have no significant correlations.

The PCA analysis demonstrated that two main components explain 76.71% of the total data variance (PC1 57.79% and PC2 18.92%). The distribution of the variables in the space of the two principal components is shown in Figure 4.

Most variables strongly contribute to PC1, as described by the PCA factor loadings in Table 5. The percentage of Cr(VI) removal (−0.9220) and eDNA (−0.6701) are negatively related to PC1, while pH (0.8756), DO (0.9579) and nitrate (0.8932) are positively related to PC1. The ORP (0.8215) and sulphate (0.7020) are positively related to PC2. 

The PCA scores applied in the CA analyses allowed us to identify five clusters, the same for both AHC and the *k-means* approaches, as shown in Figure 5.

The resulting plot demonstrates the spatial distribution of the samples in accordance with values of the Cr(VI) removal, pH, ORP, DO, eDNA, nitrate and sulphate, measured at the end of the experiment. Overall, for each treatment, most of the replicates are grouped in the same cluster, while differences between the treatments are reflected by distinct clusters. Distinct clusters mostly correspond to different treatments, with the sole exception of the ABIO and ABIO PHB series, which are grouped together. The distribution of ABIO and ABIO PHB in the first quadrant of the plot was driven by the co-occurrence of higher values of pH, DO, ORP, nitrate and sulphate, and lower values related to the percentage of the Cr(VI) removal and eDNA content. 

BIO was in the second quadrant, mainly characterized by higher pH, DO and nitrate values, and, at the same time, lower values of the ORP, sulphate, eDNA and the percentage of Cr(VI) removal.

YE was in the third quadrant of the plot, in which the samples with higher eDNA content and higher pollutant removal values, but with lower values of pH, ORP, DO, nitrate and sulphate, were grouped.

PHB, except for one replicate, was in the fourth quadrant, characterized by samples with higher values of ORP, sulphate, eDNA and Cr(VI) removal, and concurrently lower values of pH, DO and nitrate. 

ABIO YE was in the center of the plot, with intermediate characteristics. 

#### 4.3.2. MANOVA

The results of the MANOVA analysis applied to the microcosms’ properties, relating to the differently amended or unamended reactors, showed a high significance (*p*-value < 0.0001 ***) according to all three types of tests (Wilk’s, Hotelling–Lawley, Pillai, and Roy tests). 

Because the results of the multivariate test were found to be significant, the univariate ANOVA was performed to determine which of the variables and factors influenced the significance (Table 6).

The percentage of the Cr(VI) removal, pH, DO, eDNA and nitrate were found to be the main factors affecting the differences between the treatments of BIO, YE, PHB, ABIO, ABIO YE and ABIO PHB, which were statistically significant also at the maximum stringency level (*p*-value < 0.0001). On the contrary, the influence of the sulphate content was not significant.

## 5. Discussion

The pH and ORP affect chromium speciation in water, as summarized by the pH-ORP diagram, also known as the Pourbaix diagram [1]. The GWG sample had a pH close to neutrality, and the ORP was 96.4 mV (Table 2). In the microcosms, the pH was in the range of 7.0–7.9, and the redox potential was always below 300 mV throughout the experiment (Table 3). According to the Pourbaix diagram, both in the aquifer system of the study area and under the test conditions, the trivalent form of chromium is favored. This suggests the anthropogenic origin of the Cr(VI) in the study area, and consequently indicates the need to intervene to remediate the groundwater. In addition, such favorable environmental conditions represent a fundamental prerequisite for avoiding the Cr(III) re-oxidation after a remediation intervention in this site.

The results from this study excluded a remediation by natural attenuation. In fact, in the absence of an external electron donor, no mitigation of Cr(VI) pollution was observed over 28 days both in the viable and sterile reactors, BIO and ABIO (Figure 2 and Figure 3).

By contrast, the addition of yeast extract and polyhydroxybutyrate induced the pollutant removal, mediated by microbial metabolism. 

The positive outcomes achieved by using yeast extract were in accordance with previous experiments on Cr(VI) bioreduction carried out on a laboratory scale [14,38]. The yeast extract comprises valuable soluble components of yeast cells, nitrogen compounds and vitamins, which can act as reducers and sustain bacterial growth. In fact, YE showed the quickest rate, both in terms of the percentage of chromium removal and the microbial proliferation already after 2 days of incubation, achieving 100% Cr(VI) reduction after 7 days of incubation (Figure 2). In this study, the microbial proliferation was estimated by measuring the amount of eDNA, in accordance with Aoshima et al. [39], which identified a linear proportional relationship between the bacterial number obtained using 4′,6-diamino-2-phenylindole (DAPI) staining and the eDNA concentration (R^2^ = 0.96). Among the mechanisms involved in the Cr(VI) reduction (Section 1), it is possible that, in YE reactors, chromium reduction also occurs under denitrifying conditions, as previously reported by Middleton et al. [10]. In fact, the Pearson’s correlation applied to the monitored variables revealed a strong negative correlation between the Cr(VI) removal and nitrate content (n = −0.940), suggesting that the increase in the Cr(VI) removal percentage was related to a decrease in the nitrate concentration (Table 4). Moreover, the PCA demonstrated a strong correlation with the PC1 for these two parameters, but in opposite directions (Figure 4 and Table 5). Although the statistical analyses highlighted a strong correlation (n = 0.724) between the percentage of Cr(VI) removal and the eDNA (Table 4), the comparison between the performance of YE and the corresponding sterile control ABIO YE suggested that, in the presence of yeast extract, the Cr(VI) removal was the result of combined biological and chemical reduction processes. More than 60% of the Cr(VI) was chemically removed in the ABIO YE at the end of the experiment, possibly due to the organic compound oxidation (Figure 3). Among the yeast extract constituents, riboflavin and other B vitamins can act as redox mediators [40]. It is important to point out that, in the absence of viable native bacteria, the complete reduction of Cr(VI) was not achieved (Figure 3). 

The results obtained by adding polyhydroxybutyrate were consistent with the slow-release behavior of this substrate. This organic substrate, which has previously been successfully used for the in situ bioremediation of aquifers polluted by chlorinated aliphatic hydrocarbons [35], also exhibited the ability to promote the hexavalent chromium bioreduction. In PHB, the Cr(VI) removal started after 7 days of incubation, in parallel with a gradual increase of the microbial proliferation (Figure 2). In fact, bacteria can enzymatically hydrolyze polyhydroxybutyrate, subsequently converted to acetate and H_2_, which is involved in the bioreduction processes of Cr(VI) [41,42,43]. Empirical observations, i.e., naked-eye detection of the blackish layers in the solid phase of the PHB reactors, starting from 7 days until the end of the experiment, suggested that the Cr(VI) reduction could also occur by the activity of dissimilatory iron-reducing bacteria [44]. In the absence of microorganisms (ABIO PHB), polyhydroxybutyrate is an inert substrate. In fact, in the sterile controls, no reduction of the contaminant was observed (Figure 3). 

The occurrence of different mechanisms putatively involved in the pollutant removal appears also to be indicated by the spatial distribution of the samples in the cluster plot, based on the PCA scores (Figure 5). In particular, YE and PHB characterized by the bacterial removal of Cr(VI) were on the left side of the plot, in the lower and upper quadrants, respectively. BIO, ABIO and ABIO PHB, which were characterized by a lack of Cr(VI) reduction, were distributed on the right side of the plot. Interestingly, ABIO YE, which was characterized by a percentage of pollutant removal which was close to that of PHB, but not mediated by microbes, was clustered apart, in the center of the plot.

The differences observed between the microcosms differently amended and unamended were consistent with the MANOVA analysis, and were driven by the parameters of the pH, DO, percentage of Cr(VI) removal, eDNA and nitrate (Table 6).

## 6. Conclusions and Future Perspectives

The microcosm scale study carried out in this work demonstrated the high capabilities of autochthonous bacteria in promoting Cr(VI) bioreduction when subjected to biostimulation. The results showed a clear dependence of the rate of Cr(VI) reduction on the amendment used. Notably, the YE treatment was more efficient than the PHB treatment in reducing the pollutant concentration, achieving 100% and less than 10% after 7 days of incubation, respectively.

Polyhydroxybutyrate, used in this study for the first time for the purpose of Cr(VI) bioremediation, was effective, providing about 70% pollutant removal by the end of the experiment. This indicates that polyhydroxybutyrate is a suitable candidate for promoting the bacteria-mediated recovery of multi-contaminated groundwater. 

The sterile controls highlighted that the yeast extract is the only amendment that is also able to promote the abiotic Cr(VI) removal up to a maximum value of 67%, due to the organic compound’s chemical oxidation. 

Distinct trends observed in the Cr(VI) reductions in the YE and PHB reactors may be attributable to different bacterial metabolic pathways specifically induced by the amendment type. Empirical observations carried out during the viable microcosms’ operation (i.e., naked-eye detection of the changes in appearance during the solid phase) supported the hypothesis that Fe-mineral modifications, supposedly induced by microbes, are also involved in Cr(VI) metabolism. Further investigations on the oxidative state balance of Fe and Cr in the solid phase of the viable reactors will be performed using X-ray photoelectron spectroscopy (XPS).

In addition, a next-generation sequencing approach applied to the 16S rDNA extracted from the viable microcosms will provide fundamental information about the indigenous bacterial community’s structure and its possible rearrangement, triggered by biostimulation treatments. This information, together with the results presented in this study, will offer an accurate benchmark for the investigation of the microbial chromium pollution attenuation in groundwater.

## Figures and Tables

**Figure 1 ijerph-19-09622-f001:**
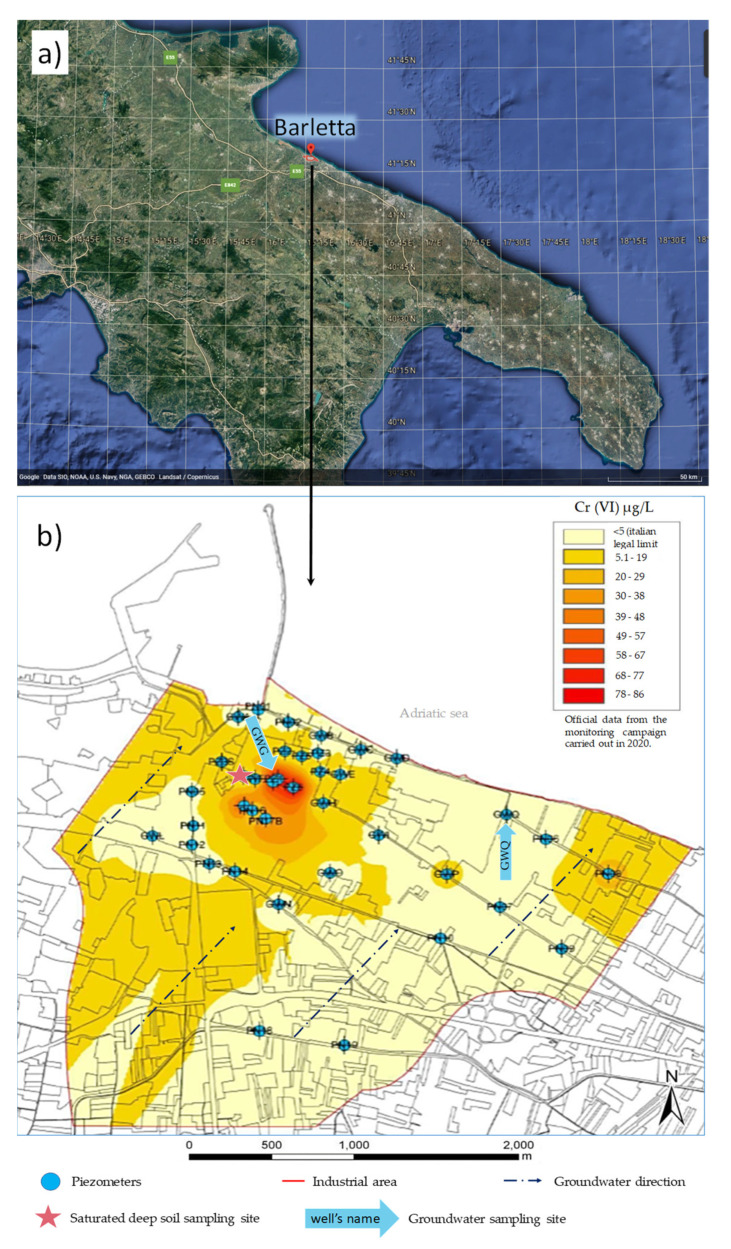
(**a**) Location of the Barletta Municipality, Apulia, Southern Italy (credit Google Earth, image Landsat/Copernicus, data SIO NOAA, U.S. Navy, NGA, GEBCO). (**b**) Sampling sites and Cr(VI) distribution in the study area according to the monitoring campaign carried out in 2020 (modified from [30]).

**Figure 2 ijerph-19-09622-f002:**
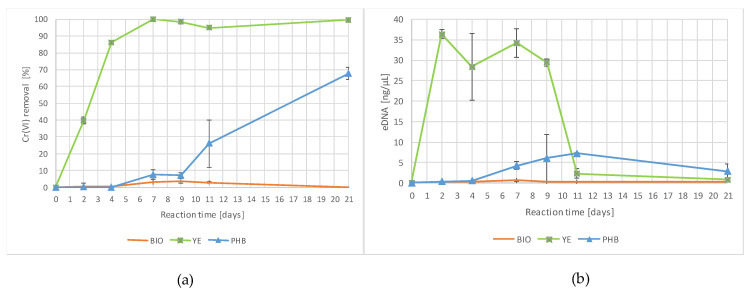
(**a**) Percentage of Cr(VI) removal and (**b**) eDNA content observed during the experiment in the viable reactors.

**Figure 3 ijerph-19-09622-f003:**
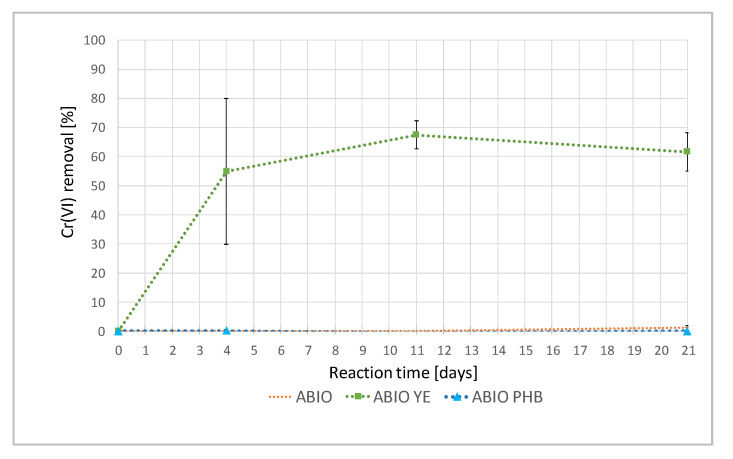
Percentage of Cr(VI) removal in the sterile reactors during the experiment.

**Figure 4 ijerph-19-09622-f004:**
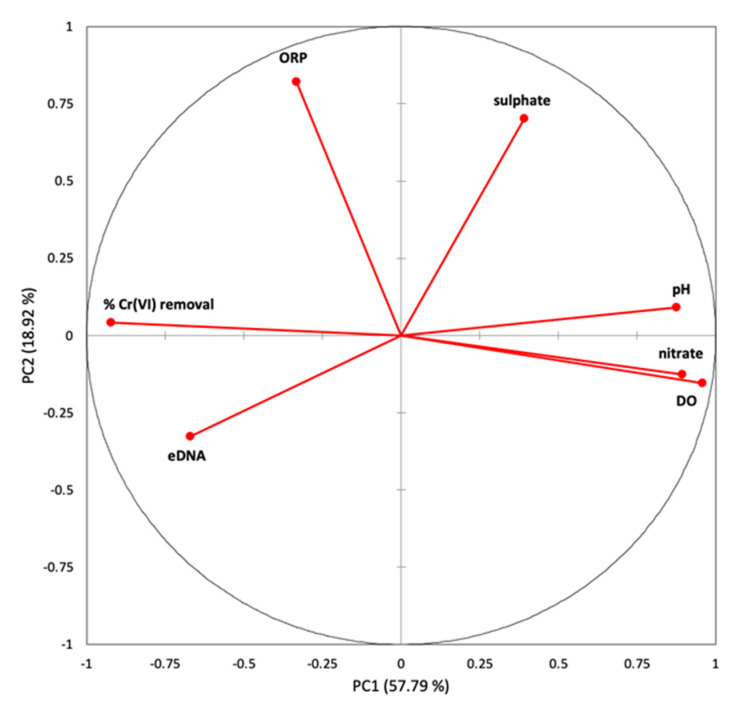
Distribution of the variables in the space of the two principal components.

**Figure 5 ijerph-19-09622-f005:**
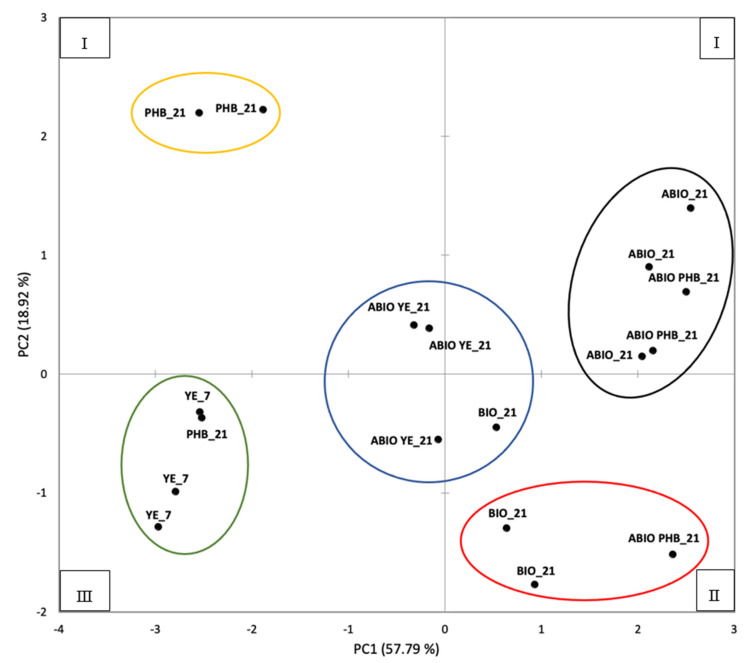
Sample clustering based on the PCA scores. The Roman numeral indicates the quadrant.

**Table 1 ijerph-19-09622-t001:** Microcosms series composition and the related number of reactors set up in this study.

	Series	Composition	Biological Replicates	Number of Observations	Reactors for Series	Dosage References
**Viable batches**	BIO	DS 50 g + GWG 200 mL	3	6	18	-
	YE	DS 50 g + GWG 200 mL + yeast extract (200 mg/L)	3	6	18	[14]
	PHB	DS 50 g + GWG 200 mL + polyhydroxybutyrate (180 mg/L)	3	6	18	[35]
	BIO blk	DS 50 g + GWQ 200 mL	1	6	6	-
	YE blk	DS 50 g + GWQ 200 mL + yeast extract (200 mg/L)	1	6	6	[14]
	PHB blk	DS 50 g + GWQ 200 mL + polyhydroxybutyrate (180 mg/L)	1	6	6	[35]
**Sterile controls**	ABIO	Autoclaved DS 50 g + autoclaved GWG 200 mL	3	3	9	-
	ABIO YE	As the same as ABIO + autoclaved yeast extract (200 mg/L)	3	3	9	[14]
	ABIO PHB	As the same as ABIO + autoclaved polyhydroxybutyrate (180 mg/L)	3	3	9	[35]
	ABIO blk	Autoclaved DS 50 g + autoclaved GWQ 200 mL	1	3	3	-
	ABIO YE blk	As the same as ABIO blk + autoclaved yeast extract (200 mg/L)	1	3	3	[14]
	ABIO PHB blk	As the same as ABIO blk + autoclaved polyhydroxybutyrate (180 mg/L)	1	3	3	[35]

**Table 2 ijerph-19-09622-t002:** Characterization of the groundwater and deep saturated soil samples.

Groundwater											
	pH	EC (µS/cm)	T (°C)	DO (ppm)	ORP (mV)	Nitrate ^a^ (mg/L)	Sulphate ^b^ (mg/L)	DOC (mg/L)	Total Fe ^b^ (µg/L)	Total Cr ^b^ (µg/L)	Cr(VI) ^b^ (µg/L)
**GWG**	6.8	3062	21.6	3.75	96.4	47.35	460.45	41.4	20	121.5	133.9
**GWQ**	7.0	1486	22.5	3.57	46.7	35.41	80.24	38.84	<20	-	<0.5
**Legal limit** ** ^a,b^ **	-	-	-	-	-	50	250	-	200	50	5
**Deep saturated soil**											
	**pH**	**Water content** **(%)**		**Texture (USDA)**	**CEC** **(meq per 100 g)**		**EC** **(µS/cm)**	**OC** **(g/kg)**	**CC** **(g/kg)**	**Total Fe (mg/kg)**	**Total Cr^b^ (mg/kg)**
**DS**	6.7	21		Sandy loam	7.7		205.4	1.5	235.7	11,161	18.27
**Legal limit** ** ^b^ **	-	-		-	-		-	-	-	-	800

^a^ Directive 91/676/EEC; ^b^ Italian legislative decree no. 152/2006. The legal limit for the total Cr in soils is referred to for the surface soils from industrial sites.

**Table 3 ijerph-19-09622-t003:** Mean values of the pH, DO, ORP, nitrate, and sulphate measured in the viable and sterile reactors at fixed monitoring times.

Series	Time (d)	pH	DO (ppm)	ORP (mV)	Nitrate (mg/L)	Sulphate (mg/L)
**Viable series**						
BIO	0	7.3 ± 0.02	5.4 ± 0.13	198 ± 9.18	50.40 ± 3.55	481.18 ± 2.61
	2	7.3 ± 0.02	5.5 ± 0.06	209 ± 6.87	90.50 ± 3.10	1104.70 ± 33.80
	4	7.3 ± 0.02	5.0 ± 0.22	194 ± 2.55	100.60 ± 4.00	1242.70 ±50.90
	7	7.3 ± 0.03	5.2 ± 0.11	226 ± 17.29	71.27 ± 6.52	919.00 ± 10.50
	9	7.3 ± 0.06	4.6 ± 0.19	225 ± 11.02	53.40 ± 1.21	593.10 ± 7.45
	11	7.3 ± 0.02	4.7 ± 0.13	233 ± 6.18	49.50 ± 1.25	563.60 ± 19.74
	21	7.2 ± 0.02	4.3 ± 0.16	199 ± 12.76	48.83 ± 9.96	617.27 ± 62.33
YE	0	7.2 ± 0.03	5.2 ± 0.06	161.7 ± 1.16	50.49 ± 3.09	535.68 ± 22.23
	2	7.2 ± 0.05	0.0 ± 0.01	14 ± 30.32	<10	710.70 ± 91.63
	4	7.1 ± 0.02	0.6 ± 0.49	54 ± 10.76	<10	890.30 ± 56.21
	7	7.1 ± 0.04	2.0 ± 0.03	202 ± 1.57	<10	608.65 ± 23.75
	9	7.1 ± 0.02	1.7 ± 0.21	176 ± 23.52	<10	608.83 ± 20.65
	11	7.0 ± 0.02	1.8 ± 0.12	226 ± 12.28	<10	592.83 ± 25.53
	21	7.0 ± 0.02	1.2 ± 0.17	189 ± 7.99	<10	589.43 ± 14.81
PHB	0	7.3 ± 0.01	5.1 ± 0.07	195 ± 4.28	50.30 ± 4.23	501.91 ± 5.22
	2	7.3 ± 0.02	5.1 ± 0.09	208 ± 1.47	50.30 ± 14.27	446.13 ± 16.70
	4	7.3 ± 0.02	5.0 ± 0.14	175 ± 12.90	71.35 ± 0.55	876.50 ± 30.70
	7	7.2 ± 0.02	3.4 ± 0.14	201 ± 2.32	22.75 ± 0.85	953.70 ± 23.70
	9	7.1 ± 0.04	2.8 ± 0.32	216 ± 10	23.03 ± 10.95	605.50 ± 2.20
	11	7.1 ± 0.02	2.8 ± 0.56	220 ± 3.73	<10	613.17 ± 37.49
	21	7.0 ± 0.03	1.1 ± 0.09	237 ± 17.77	<10	615.80 ± 55.87
**Sterile series**						
ABIO	0	7.9 ± 0.05	5.3 ± 0.25	188 ± 12.14	56.25 ± 3.45	635.95 ± 46.55
	4	7.6 ± 0.30	5.5 ± 0.09	201 ± 14.20	56.00 ± 2.26	648.93 ± 24.45
	11	7.9 ± 0.07	5.5 ± 0.11	216 ± 12.69	57.90 ± 2.17	681.50 ± 28.72
	21	7.7 ± 0.04	5.3 ± 0.13	214 ± 5.27	57.00 ± 4.22	655.70 ± 24.76
ABIO YE	0	7.9 ± 0.02	5.2 ± 0.01	125 ± 3.35	59.30 ± 5.35	695.85 ± 5.85
	4	7.4 ± 0.04	2.9 ± 0.06	122 ± 5.33	<10	634.00 ± 6.89
	11	7.5 ± 0.01	4.7 ± 0.57	118 ± 2.01	<10	673.67 ± 27.84
	21	7.4 ± 0.03	4.1 ± 0.26	198 ± 9.99	<10	634.93 ± 7.07
ABIO PHB	0	7.9 ± 0.03	5.1 ± 0.08	181 ± 9.60	53.25 ± 1.65	599.15 ± 14.15
	4	7.9 ± 0.05	5.5 ± 0.14	181 ± 3.50	53.45 ± 0.25	611.40 ± 0.20
	11	7.8 ± 0.02	5.6 ± 0.11	233 ± 2.00	62.67 ± 3.03	735.43 ± 31.48
	21	7.7 ± 0.04	5.5 ± 0.13	196 ± 21.13	55.87 ± 2.73	640.50 ± 24.35

**Table 4 ijerph-19-09622-t004:** Correlation between the monitored parameters (Pearson’s coefficient, n). Strongest correlations (n > |0.5|) are in bold.

Variables	% Cr(VI) Removal	pH	ORP	DO	eDNA	Nitrate
pH	**−0.646**					
ORP	0.202	−0.315				
DO	**−0.828**	**0.886**	−0.487			
eDNA	**0.724**	−0.411	−0.092	**−0.544**		
nitrate	**−0.940**	**0.690**	−0.258	**0.810**	−0.481	
sulphate	−0.146	**0.577**	0.212	0.326	−0.240	0.166

**Table 5 ijerph-19-09622-t005:** Factor loadings of the first two main components of the PCA.

Variables	PC1	PC2
% Cr(VI) removal	−0.9220	0.0429
pH	0.8756	0.0909
ORP	−0.3334	0.8215
DO	0.9579	−0.1543
eDNA	−0.6701	−0.3272
nitrate	0.8932	−0.1254
sulphate	0.3907	0.7020

**Table 6 ijerph-19-09622-t006:** *p*-values of the ANOVA for the microcosms’ monitored parameters.

	Cr (VI) Removal	pH	ORP	DO	eDNA	Nitrate	Sulphate
significance	***	***	*	***	***	***	ns
*p*-value	1.57 × 10^−16^	9.8181 × 10^−12^	0.0195	4.8412 × 10^−13^	1.2035 × 10^−12^	5.7529 × 10^−10^	0.0990

* = the difference is significant at *p* < 0.05; *** = the difference is significant at *p* < 0.001; ns = the difference is not significant.

## Data Availability

The data presented in this study are available on request from the corresponding author.

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
