# Peer review of "Enhanced Natural Attenuation of Groundwater Cr(VI) Pollution Using Electron Donors: Yeast Extract vs. Polyhydroxybutyrate"

_ijerph, 2022, doi:10.3390/ijerph19159622_

Round 1

Reviewer 1 Report

The manuscript entitled "Enhanced natural attenuation of groundwater Cr(VI) pollution, using electron donors: yeast extract vs polyhydroxybutyrate" aims to  evaluate the feasibility of a natural attenuation, or alternatively an enhanced natural attenuation strategy, for recovering Cr(VI) polluted groundwater. 

 The purpose of this study is clear, the method is appropriate, and it has a certain practical significance.  

However, there are still some questions to be considered as follows:

Specific comments:

1. The introduction of the research background is not detailed enough. The author should clearly point out the current research status and the existing problems. 

2. Is there any existing relevant research in the study area? Please also make a brief introduction.

3. Please briefly introduce the principle of sample point layout. What equipment is used for depth sampling (i.e. 14m)? Is the depth of each well 20m? How many samples were finally obtained?

4. What is the recovery rate of sample determination? 

Reviewer 2 Report

The manuscript entitled "Enhanced natural attenuation of groundwater Cr(VI) 2 pollution, using electron donors: yeast extract vs 3 polyhydroxybutyrate" analyses bioremediation of Cr-contaminated groundwater and soil using yeast extract or polyhydroxybutyrate. The work is interesting and clearly presented, I would only have some minor observations. 

Why did the authors choose to use polyhydroxybutyrate? Motivation of this decision is not clearly stated in the text. It should be mentioned in the Introduction section.

Also, differences between the effect of the yeast extract should be stated more clearly both in the Discussion and in the Conclusion sections. 
